# The impact of the SARS-COV2 infection on the disorder of consciousness rehabilitation unit

**Silvia Marino, Rosella Ciurleo** *, **Antonino Todaro, Antonella Alagna, Anna Lisa Logiudice, Francesco Corallo, Caterina Formica, Carmela Rifici, Patrizia Pollicino, Fabrizia Caminiti, Elisabetta Morini, Placido Bramanti**

IRCCS Centro Neurolesi "Bonino Pulejo", Messina, Italy

* rossella.ciurleo@irccsme.it

## Abstract

### Background and objective

Disorders of consciousness include coma (cannot be aroused, eye remain closed), vegetative state—VS (can appear to be awake, but unable to purposefully interact) and minimally conscious state—MCS (minimal but definite awareness). The objective of this study is to assess the impact of the SARS-CoV-2 infection on the Disorder of Consciousness (DOC) Rehabilitation Unit.

### Methods

This is a retrospective, longitudinal, descriptive, observational, pilot study. We consecutively enrolled 18 patients (age range: 40–72 years, 9 females and 9 males), from three to five months after a brain injury. They were grouped into VS (n = 8) and MCS (n = 10). A confirmed case of COVID-19 was defined as a positive result on high-throughput sequencing or real-time reverse-transcription polymerase chain reaction analysis of throat swab specimens. We collected data of lung Computed Tomography (CT) and laboratory exams. DOC patients who were positive for SARS-CoV-2 were classified into severe and no severe infected group, according to the American Thoracic Society guidelines.

### Results

A total of 18 hospitalized patients with (16) and without confirmed (2) SARS-CoV-2 infection were included in the analysis. After one month, a follow-up clinical evaluation reported that one patient died, one patient was transferred from Covid Unit to Emergency Unit and 3 patients were resulted negative to double swab and they returned to Rehabilitative Unit. Significant differences were reported about hypertension, cardiac disease and respiratory problems between the patients with severe infection and patients without severe infection (P< 0.001). The laboratory findings, such as blood cell counts ($P < 0.001$), C-reactive protein, D-dimer, potassium and vitamin D levels, seemed to be considered as useful prognostic predictors.

**Funding:** Supported by Italian Health Minister (GR-2013-02359341).

**Competing interests:** The authors have declared that no competing interests exist.

## Conclusions

To our knowledge, this is the first longitudinal study on a sample of chronic DOC patients affected by SARS-CoV-2. This study may offer important new clinical information on COVID-19 for management of DOC patients. Our findings showed that for the subjects with severe infection due to COVID-19, rapid clinical deterioration or worsening could be associated with clinical and laboratory findings, which could contribute to high mortality rate. During the COVID-19 epidemic period, the clinicians should consider all the reported risk factors to avoid delayed diagnosis or misdiagnosis and to prevent the infection transmission in DOC Rehabilitation Unit.

## Introduction

On March 11, 2020, the World Health Organization (WHO) declared the severe acute respiratory syndrome coronavirus 2 (SARS-CoV-2) outbreak a pandemic due to the dramatic increasing number of cases outside China [1, 2].

In Italy, on February 2020 the emergence of the COVID-19 epidemic first in Lombardy, and then in other regions, determined the need to implement containment measures for a phenomenon that stressed national healthcare system [3].

It is known that the coronaviruses can adapt very quickly and cross the species barrier, such as with SARS-CoV and Middle East respiratory syndrome CoV (MERS-CoV). In humans, coronavirus infection may often lead to severe clinical symptoms and high mortality. As for COVID-19, several studies have described typical clinical manifestations including fever, cough, anosmia, ageusia, diarrhea, fatigue and other symptoms. COVID-19 also presented characteristic laboratory findings and lung computed tomography (CT) abnormalities.

In a retrospective, observational study [4], the authors reported neurologic manifestations in a COVID-19 Chinese population.

Patients with severe disability caused by a neurological disease often have different comorbidities and many risk factors for poor outcome in SARS-COV2 infection.

Disorders of Consciousness (DOC) include patients in coma, vegetative/unresponsive wakefulness syndrome (VS/UWS) and minimally consciousness state (MCS) [5, 6].

DOC patients are considered as a particular and delicate population that presents important pre-existing impairment in central nervous, cardiac and respiratory systems.

This particular clinical population often needs long-term treatments and devices, such as tracheostomy tube, central venous catheters, nasogastric or gastrostomy tube and indwelling catheters, which highly increase the comorbidity burden.

In addition, a significant high rate of poor outcomes and mortality should be expected in DOC patients who become infected with SARS-CoV-2.

To date, no data are available on the impact of the SARS-CoV-2 infection on survival of chronic DOC patients in Rehabilitation DOC Unit. In fact, to our knowledge, this is the first longitudinal study on a sample of DOC patients affected by SARS-CoV-2.

## Methods

### Study design

This is a retrospective, longitudinal, descriptive, observational pilot study. All patients were consecutively assessed from 15 February to 26 March 2020 in post-acute rehabilitation DOC Unit of IRCCS Centro Neurolesi "Bonino-Pulejo" of Messina (Italy).

**Table 1. Clinical and demographic details of DOC patients.**

| Patient | clinical diagnosis | gender/age | etiology | Lesions (CT or MRI) | Month since injury | CRS-R sub-scores | CRS-R total scores |
|---|---|---|---|---|---|---|---|
| 01 | MCS | F/42 | Trauma | Bilateral frontal, right parietal, lobe lesions | 4.1 | 2 1 3 1 0 2 | 8 |
| 02 | MCS | M/67 | Hemorrhage | Bilateral temporal lobe lesions | 3.0 | 1 1 3 0 0 2 | 7 |
| 03 | MCS | F/72 | Hemorrhage | Right frontal, temporal lobe and brain stem lesions | 4.9 | 1 2 3 0 0 2 | 8 |
| 04 | MCS | M/41 | Trauma | Bilateral frontal lobe lesions | 3.2 | 1 1 4 0 0 2 | 8 |
| 05 | MCS | F/47 | Anoxia | Diffuse demyelination | 3.1 | 2 3 2 1 0 1 | 9 |
| 06 | MCS | M/52 | Hemorrhage | Brain stem and cerebellum lesions | 4.8 | 1 1 3 0 0 3 | 8 |
| 07 | MCS | M/71 | Hemorrhage | Brain stem lesions | 4.0 | 1 3 4 0 1 1 | 10 |
| 08 | MCS | F/42 | Hemorrhage | Left parietal lobe lesions | 4.8 | 2 1 3 0 0 1 | 7 |
| 09 | MCS | M/65 | Trauma | Bilateral frontal lobe and left parietal lobe lesions | 4.2 | 2 1 3 1 0 1 | 8 |
| 10 | VS | M/62 | Anoxia | Diffuse demyelination | 5.0 | 1 0 1 0 0 1 | 3 |
| 11 | VS | M/38 | Anoxia | Diffuse demyelination | 5.0 | 1 0 1 1 0 1 | 4 |
| 12 | VS | M/67 | Trauma | Right frontal and parietal lobe lesions | 3.4 | 1 1 1 0 0 1 | 4 |
| 13 | VS | F/72 | Hemorrhage | Right temporal, and occipital lobe lesions | 3.7 | 1 0 1 0 0 2 | 4 |
| 14 | VS | F/68 | Trauma | Bilateral parietal and temporal lesions | 4.8 | 1 1 1 0 0 2 | 5 |
| 15 | VS | F/63 | Anoxia | Diffuse demyelination | 4.1 | 0 0 1 0 0 2 | 3 |
| 16 | VS | M/55 | Hemorrhage | Left frontal and temporal lobe lesions | 3.7 | 1 1 1 0 0 1 | 4 |
| 17 | VS | F/64 | Trauma | Right parietal and temporal lobe lesions | 3.4 | 1 0 1 0 0 2 | 4 |
| 18 | VS | F/40 | Anoxia | Diffuse demyelination | 3.3 | 1 1 0 0 0 1 | 3 |

CRS-R = Coma Recovery Scale-Revised; Six subscales score of CRS-R indicating the assessment of auditory, visual, motor, verbal, communication functions and arousal. DOC = Disorder of Consciousness; MCS = minimally consciousness state; VS = vegetative syndrome.

## Participants

We studied 18 patients (age range: 40–72 years, 9 females and 9 males) from three to five months after a brain injury (see Table 1 for clinical and demographic details). By using validated behavioral scales such as Revised Coma Recovery Scale (CRS-R), the patients were grouped into VS (n = 8) and MCS (n = 10). CRS-R is unique in establishing a diagnosis and outcome directly from the examination findings. The basic structure of the CRS-R is similar to the Glasgow Coma Scale (GCS), but its subscales (auditory, visual, motor, oromotor/verbal, communication, and arousal) are much more detailed, targeting more subtle signs of recovery of consciousness [7]. A confirmed case of COVID-19 was defined as a positive result on high-throughput sequencing or real-time reverse-transcription polymerase chain reaction analysis of throat swab specimens.

## Interventions

We collected data of lung Computed Tomography (CT), laboratory exams (blood cell count, blood chemical analysis, coagulation testing, liver and renal function testing, C-reactive protein, vitamin D serum level, glycemic state control, electrolytes) and other factors, such as Body Mass Index (BMI) and steroid regime use.

## Comparison

We compared all collected data between patients with severe and non severe infection, according to the American Thoracic Society guidelines.

### Ethics

No ethical committee approval was necessary according to national regulations because this was a retrospective analysis of routinely collected anonymized clinical data. However, the written informed consent was obtained from the legal guardian of all patients.

### Statistical analysis

Mean and standard deviations (SD) were used for normally distributed data and median and range for data that were not normally distributed. Continuous variables were compared by using Wilcoxon rank sum test. Proportions for categorical variables were compared using $\chi^2$ test. Because of the small sample size, ordinal, interval and ratio measures are presented as median and range.

All statistical analyses were performed using R (version 3.3). The level of significance was set at a $P$ less than 0.05.

## Results

A total of 18 hospitalized patients with (16) and without confirmed (2) SARS-CoV-2 infection were included in the analysis. Of these patients, 15 presented at least one of the following disorders: hypertension (11 patients), diabetes (8 patients), cardiac disease (5 patients). The most common symptoms at onset of COVID-19 disease were fever (15 patients) and respiratory problems (7 patients).

According to the American Thoracic Society guidelines [8], among the 16 patients having confirmed SARS-CoV-2 infection, 10 patients had severe infection and 6 patients had non severe infection. The patients with severe infection were significantly older (mean age, 61.1 years vs 48.4 years; $P$ <0 .001) and presented other disorders, such as hypertension (10 vs 5), and other typical COVID-19 symptoms such as fever (10 vs 5) and respiratory problems (7 vs 5), if compared with no severe infection subjects. Sixteen patients were diagnosed as having COVID-19 by a positive SARS-CoV-2 nucleic acid detection, and then they were transferred to a specific Covid Unit. Lung CT showed a typical interstitial pneumonia picture, associated to ground-glass opacity. Patients with severe infection had more increased inflammatory response, including higher white blood cell counts, lower lymphocyte and neutrophil counts, and increased C-reactive protein levels compared with the patients without severe infection ($P < 0.001$). In addition lower potassium levels in patients with severe infection were found (P<0.001).

The patients with severe infection presented also higher D-dimer levels than patients without severe infection ($P < 0.001$), probably due to a consumptive coagulation system. In addition, patients with severe infection presented multiple organ involvement, such as kidney (increased creatinine levels—P <0.01) and liver (increased alanine aminotransferase—P <0.03 and aspartate aminotransferase levels—P <0.01). All SARS-CoV-2 patients with infection had a low vitamin D serum level (see Table 2).

In addition, all patients presented a low BMI (median 17 for COVID+ and median 19.5 for COVID-).

After one month, a follow-up clinical evaluation reported that one patient had died, one patient was transferred from Covid Unit to Emergency Unit and 3 patients tested negative to double swab and returned to Rehabilitative Unit.

The first patient died for cardiac arrest. He was a 60 years old post-anoxic heart disease VS patient without severe infection.

**Table 2. Laboratory findings of DOC COVID-19 and not COVID-19.**

| Laboratory findings | Total Covid + (n.16) | Covid + severe (n.10) | Median (range) | Covid—(n.2) | P |
|---|---|---|---|---|---|
| | | | Covid + no severe (n.6) | | |
| Count, x 10⁹/L | | | | | |
| White blood cell | 5.4 (0.9–19.2) | 6.3 (0.9–19.2) | 4.9 (2.3–15.4) | 4.0 (2.9–5.1) | <0.005 |
| Neutrophil | 3.1 (0.3–17.2) | 2.9 (0.3–17.2) | 3.4 (2.9–15.4) | 8.3 (4.2–12.5) | <0.001 |
| Lymphocyte | 1.0 (0.4–2.3) | 0.8 (0.4–2.3) | 1.5 (0.9–2.1) | 1.9 (1.5–2.3) | <0.001 |
| C-reactive proteine, mg/L | 36.2 (1.2–180.3) | 55.2 (2.2–180.3) | 10.1 (1.2–46.3) | 28.2 (1.5–55.0) | <0.001 |
| D-dimer, mg/L | 0.9 (0.5–18.3) | 1.1 (0.5–18.3) | 0.8 (1.1–9.7) | 0.8 (0.8–0.9) | <0.001 |
| Aminotransferase, U/L | | | | | |
| Alanine | 33.2 (12.0–877.2) | 44.5 (12.0–877.2) | 21.3 (15.3–289.2) | 45.5 (33.0–58.0) | <0.03 |
| Aspartate | 29.3 (9.0–933.1) | 44.3 (9.0-.933–1) | 25.1 (12.3–198.5) | 28.0 (12.0–44.0) | <0.01 |
| Creatinine, μmol/L | 55.3 (31.2–833.2) | 66.2 (31.2–833-2) | 52.5 (22.1–125.4) | 29.0 (25.0–33.0) | <0.01 |
| Vitamin D ng/ml | 7.1 (6.5–28.3) | 6.8 (6.5–28.3) | 11.2 (10.1–55.3) | 39.5 (35.0-44-0) | <0.001 |
| Potassium mmol/L | 1.85 (1,2–3,9) | 1.6 (1.2–1.9) | 2,9 (2.3–3.9) | 3.3 (3.1–3.5) | <0.001 |
| Sodium mmol/L | 139 (135–143) | 137 (135–139) | 142 (140–143) | 141.5 (141–142) | <0.007 |
| Calcium mmol/L | 7.9 (6.9–8.9) | 7.3 (6.9–7.9) | 8,3 (8.1–8.9) | 8.1 (7.9–8.3) | <0.007 |
| Glycemia mg/dl | 178 (113–267) | 233 (211–267) | 189 (123–234) | 133 (123–143) | <0.007 |

DOC = Disorder of Consciousness; COVID-19 = Coronavirus Disease 2019.

The second patient was transferred from Covid Unit to Emergency Unit for respiratory insufficiency. He was a young post-anoxic VS patient, affected by Brugada syndrome, with severe infection and serious multiple organ involvement.

The three patients who returned to Rehabilitative DOC Unit, after a double negative swab, were MCS patients, without severe infection (one 40 years old post-traumatic patient, one post-hemorrhagic middle age patient and one 40 years old post-hemorrhagic patient). In addition, they presented lower-than-normal vitamin D serum and potassium levels, but higher than those of patients with severe infections (P < 0.01).

After one month, a follow-up clinical evaluation reported that the remaining 11 patients (9 with severe and 2 without severe infection), were clinically stable and hospitalized at the Covid Unit, as they were still positive to control swabs.

Finally, 2/18 VS patients (one post-hemorrhagic middle age patient and one 40 years old post-anoxic patient) were negative to a double swab. The female showed: 1) fever; 2) typical higher white blood cell and neutrophil counts; 3) lower lymphocyte counts; 4) increased C-reactive protein levels. These findings were similar to SARS-CoV-2 positive patients with severe infection, even if she showed a no typical COVID-19 lung CT. The male patient presented: 1) no fever; 2) normal white blood cell counts; 3) no typical COVID-19 lung CT. Finally, both patients had normal D-dimer and vitamin D serum levels (see Table 2). Glycemic state control was abnormal in all patients (see Table 2), but no significant correlation with level of infection severity was found (P< 0.007).

In addition, all patient with confirmed SARS-CoV-2 infection were treated with dexamethasone 6 mg IV once daily plus standard of care for up to 10 days or until hospital discharge.

## Discussion

To our knowledge, this is the first longitudinal report of DOC patients with SARS-CoV-2 hospitalized in a Rehabilitation Unit.

Our sample is to be considered as very particular population, because DOC patients very often present several neurological, cardiological and pulmonary problems. Indeed, they are more likely to contract infections than other neurological patients.

The main results of the study showed that DOC patients are unfortunately a population susceptible to SARS-CoV-2 infection, especially if there is an outbreak of infection. In fact, seriously compromised systems, tracheostomy tube and particular radiological and laboratory settings could be responsible for a high susceptibility to contract the infection.

Specifically, of 18 patients included in this study, 16 were positive for COVID-19 (10 had severe infection and 6 had no severe infection) and 2 were negative. The patients with severe infection were older than those without severe infection and presented hypertension and symptoms such as fever and respiratory problems. Our patients affected by severe infection had all tracheotomy: this aspect could be favoring the infection and the gravity of pulmonary findings. Post-acute neurological pneumonia is associated with old age, history of diseases, neurological impairment and disease severity [9], and it is known that lung infections in these patients often lead to poor prognoses. In addition, dysphagia was present in all patients with severe infection and it is known to be one of the most common complications in post-acute neurological patients and is also a significant risk factor for lung infections in DOC patients.

Cardiological problems could also be considered as a poor prognostic factor. In fact, all patients with severe infection presented one or more comorbidity such as hypertension and other cardiac problems.

In addition, the 2 VS subjects who had a worse prognosis showed a pre-existing serious cardiological disease. The first patient died for cardiac arrest. He was a post-anoxic heart disease VS patient without severe infection. The second patient was transferred from Covid Unit to Emergency Unit for respiratory insufficiency. He was a post-anoxic VS patient affected by Brugada syndrome, with severe infection and serious multiple organ serum involvement.

The three patients who returned to our Rehabilitative Unit, after a double negative swab, did not show cardiac problems.

Another important aspect is the lymphocyte count. COVID-19 is a systemic infection with a significant impact on the hematopoietic system and hemostasis. Lymphopenia may be considered as a cardinal laboratory finding, with an important prognostic role. Neutrophil/lymphocyte ratio and peak platelet/lymphocyte ratio may also have prognostic value in determining severe cases. Biomarkers, such high serum procalcitonin and ferritin have also emerged as poor prognostic factors. Furthermore, blood hypercoagulability is common among the hospitalized COVID-19 patients. Elevated D-Dimer levels are consistently reported, whereas their gradual increase during disease course is particularly associated with disease worsening [10]. We found that lymphocyte count was lower in patients with severe infection. This phenomenon could be indicative of the immunosuppression in patients with severe COVID-19 disease, even if our sample including immunosuppressed subjects for pre-existing clinical conditions. Moreover, we found that patients with severe infection had higher D-dimer levels if compared with no severe infection group. The VS patient who died, had a very low lymphocyte count and a very high level of D-dimer if compared to survival VS patients with severe infection.

Also vitamin D could be an important potential protective factor for SARS-CoV-2 infection.

Vitamin D plays a key role in calcium metabolism, and its involvement in the immune response [11], its protective effect on respiratory tract infections [12] and its immunomodulatory effect in patients with pneumonia [13] have been described. However, studies of hypovitaminosis D as a risk factor for severe complications in patients with pneumonia are conflicting [14], and a meta-analysis showed that hypovitaminosis D appears to be associated with an

increased risk of adverse events in patients with pneumonia, although the molecular mechanisms related to the role of vitamin D against infections need further investigation [15].

Evidence supporting the role of vitamin D in reducing the risk of COVID-19 indicate that: a) the outbreak occurs in winter, a time when 25-hydroxyvitamin D (25(OH)D) concentrations are lowest; b) the number of cases in the Southern Hemisphere near the end of summer are low; c) vitamin D deficiency contributes to acute respiratory distress syndrome; d) case-fatality rates increase with age and with chronic disease comorbidity, both of which are associated with lower 25(OH)D concentration [16].

In our sample, the 3 patients who returned in our Rehabilitation Unit presented a vitamin D serum levels higher than those of subjects with severe infection. In addition, the 2 VS patients who were double negative, had normal vitamin D serum level.

The small number of studies investigating the immunomodulatory effects of vitamin D metabolites in respiratory virus-infected epithelial cells showed that the effects of vitamin D metabolites on the expression and secretion of pro-inflammatory cytokines and chemokines varied among the pathogens and suggested that they might be more effective against some pathogens than others [17].

In addition glycemic state control was abnormal in all patients (see Table 2), but no significant correlation with level of infection severity was found (P< 0.007). This was probably related to the fact that patients were treated with prolonged steroid therapy.

Finally all patients showed a low BMI (median 17 for COVID+ and median 19.5 for COVD-), probably related to: a) prolonged bed rest; b) diffuse muscular hypotrophy; c) parenteral or enteral nutrition.

Another important aspect is the role of the Neurorehabilitation Units, such as those in which our DOC patients were admitted. These Units should admit only patients with sub-acute neurological disorders negative for Covid-19 infection in order to facilitate prompt availability of intensive care unit.

Our sample of DOC patients demonstrated the vulnerability of this Unit and the clinical precariousness of patients themselves. The greatest difficulty in applying these indications in the field of rehabilitation is related to the need to find the right balance between the provision of services useful to the patient and the reduction of the risk of spreading the virus.

The implementation of rehabilitative activities in hospital stays, and in services in general, can only be continued in compliance with the needs of the patients and the protection of the health of all staff, as the healthcare activities require close contact with the patient, or with the production of aerosols and secretions (as for respiratory rehabilitative interventions especially in tracheostomised management patients).

This study has, however, some limitations. First, only 18 patients were studied, even if it is important to underline that all studies that include DOC subjects do not report a very high sample size, due to a difficulty to recruit this category of patients. However, it is a first longitudinal study on DOC population infected by SARS-CoV-2.

Second, during the outbreak period of COVID-19, neuroimaging evaluation, such as magnetic resonance imaging, was avoided to reduce the risk of cross infection. Thus, we have no neuroimaging data about neurological outcomes that might worsen under infection conditions. Indeed, increasing evidence showed that coronaviruses are not always confined to the respiratory tract and that they may also invade the central nervous system, through the olfactory system for example, inducing neurological outcomes [18]. Therefore, it is not excluded that SARS-CoV-2 behave in this respect like other coronaviruses. Third, this is a retrospective, longitudinal, descriptive, observational pilot study: not having a control group or not having randomized patients, is certainly another limitation. But the particular category of patients, so

difficult to recruit, and the possibility of having DOC patients affected by SARS-CoV-2 make these results unique.

In fact, disorder of consciousness represents a unique clinical condition, which poses problems that are less common in other diseases. Because of their distinctive characteristics, we preferred not to mix DOC patients with other neurological patients, even if this decision provided in a small sample size.

In conclusion, this study may offer important new clinical information on COVID-19 for management of DOC patients in Rehabilitation Unit. Our findings showed that for subjects with severe infection due to COVID-19, rapid clinical deterioration or worsening could be associated with clinical and laboratory findings which could contribute to high mortality rate. Moreover, from our results, some comorbidities seem to have a predictive role. Finally, during the epidemic period of COVID-19, clinicians should consider all the reported risk factors to avoid delayed diagnosis or misdiagnosis and to prevent the infection transmission in DOC Rehabilitation Unit.

## Supporting information

**S1 Data.**
(XLSX)

## Author Contributions

**Conceptualization:** Silvia Marino.

**Data curation:** Rosella Ciurleo.

**Formal analysis:** Antonino Todaro.

**Investigation:** Antonella Alagna.

**Methodology:** Anna Lisa Logiudice.

**Project administration:** Francesco Corallo.

**Resources:** Caterina Formica.

**Supervision:** Carmela Rifici.

**Validation:** Placido Bramanti.

**Writing – original draft:** Patrizia Pollicino, Fabrizia Caminiti.

**Writing – review & editing:** Elisabetta Morini.

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
