## [Decision Letter · Decision Letter 0]

25 Sep 2020

PONE-D-20-26757

The impact of the SARS-COV2 infection on the Disorder of Consciousness Rehabilitation Unit

PLOS ONE

Dear Dr. Ciurleo,

Thank you for submitting your manuscript to PLOS ONE. After careful consideration, we feel that it has merit but does not fully meet PLOS ONE’s publication criteria as it currently stands. Therefore, we invite you to submit a revised version of the manuscript that addresses the points raised during the review process.

ACADEMIC EDITOR: Please see comments by the reviewers below. In your revised manuscript, kindly provide point by point response to their queries. I look forward to reviewing your revised manuscript

We look forward to receiving your revised manuscript.

Kind regards,

Muhammad Adrish

Academic Editor

PLOS ONE

Journal Requirements:

Reviewers' comments:

Reviewer's Responses to Questions

**Comments to the Author**

1. Is the manuscript technically sound, and do the data support the conclusions?

Reviewer #1: Yes

Reviewer #2: Partly

2. Has the statistical analysis been performed appropriately and rigorously? 

Reviewer #1: I Don't Know

Reviewer #2: Yes

3. Have the authors made all data underlying the findings in their manuscript fully available?

Reviewer #1: Yes

Reviewer #2: Yes

4. Is the manuscript presented in an intelligible fashion and written in standard English?

Reviewer #1: No

Reviewer #2: No

5. Review Comments to the Author

Reviewer #1: I thank the authors for their challenging work. However, the paper needs lexical review, for example the choice of the words “ cardiopathic “ in page 5, “ always “ in page 6 and “ typology “ in page 8 is less familiar as the usual scientific lexicon.

Reviewer #2: - Overall, the research question is interesting however, major comments should be addressed and clarified.

- The readability and syntax of the manuscript will be substantially improved if it is reviewed by a formal translation agency or by a colleague whose first language is English.

Abstract:

- Short background on the DOC.

- Define VS and MCS.

- The design of the study should be described in the methods.

- The criteria of the subjects should be demonstrated e.g. age, genders.

- Did the author make two classifications in this study? VS and MCS, then severe and non-severe. Please, clarify

- Methods need more information.

- The size is very small to provide a conclusion.

- Change 16/18 to 16 and 2/18 to 2.

- Important differential results with p-values should be demonstrated.

- The conclusion should more concise to be related to the objective of the study.

Introduction:

- The introduction needs to be analyzed and rewritten.

- More information should be clarified about the DOC and its relation to SARS-CoV-2.

- The objective of the study is very important however the importance of the study still need more clarification in the introduction section.

- What was the hypothesis of the study.

Methods:

- What is the design of the study?

- What were the inclusion and exclusion of the study to select the participants?

- Did the author calculate the sample size and power of the study?

- Methods section is very poor and needs more information and description.

- Statistical analysis: What is the statistical method used to assess the normal distribution of the collected data. "The significance threshold" is inadequate expression. Please, change it to "the level of significance".

Results:

- Results need to provide answers to the questions raised/researchable problem.

- Results need to follow accuracy, brevity, and clarity.

- Kindly frame it along the following elements of results.

Discussions:

- The introductory paragraph should demonstrate the main findings of the study.

- The findings should be compared with previous or related studies.

- Implications of the study should be explained.

- The main limitation of the study design was not demonstrated.

- Briefly, the discussion section need major revisions.

6. PLOS authors have the option to publish the peer review history of their article (what does this mean?). If published, this will include your full peer review and any attached files.

Reviewer #1: **Yes: **MM

Reviewer #2: **Yes: **Walid Kamal Abdelbasset

---

## [Author Response · Author response to Decision Letter 0]

4 Mar 2021

Reviewer #1: 

We thank the reviewer for the kind and appreciative words. We revised the paper, as suggested.

Reviewer #2: 

We thank the reviewer for the time spent to revise the manuscript. We revised the paper, as suggested.

1) We improved English language English.

2) Abstract:

- We performed a short background on the DOC.

- We defined VS and MCS.

- We described the design of the study in the methods.

- We demonstrated the criteria of the subjects.

- We clarified the two classifications: VS and MCS, then severe and non-severe. 

-We improved “Methods section”.

- Unfortunately, the sample population cannot be expanded, but 18 DOC patients, affected by SARS-COV2, are a considerable number and never reported in the literature. 

- We changed Change 16/18 to 16 and 2/18 to 2.

- We reported the most important differential results with p-values.

- We revised the conclusion, as suggested.

3) Introduction:

- We revised the introduction, as suggested.

- No data, to date, were reported about DOC and relation to SARS-CoV-2: this is the first study. In addition this is a retrospective, longitudinal, observational study, that showed the impact, in a descriptive way, of a sudden SARS-COV2 infection. No power analysis was performed, because all patients of DOC Unit, were consecutively enrolled.

- We revised the report of objective of the in the introduction section. 

- We better explained the hypothesis of the study.

4) Methods:

- Design of the study: this is a retrospective, longitudinal, descriptive, observational study. It was reported.

- We included in the study all patients admitted to our DOC rehabilitative unit, at the time the SARS-COV2 outbreak broke out. For this reason, no particular inclusion and/or exclusion were applied.

- The response is very similar to the previous point: we included in the study all patients admitted to our DOC rehabilitative unit, at the time the SARS-COV2 outbreak broke out. For this reason, no sample size and power of the study were calculated.

- We better described “methods section”.

- We better described “statistical analysis section”.

5) Results:

- We better described “results section”.

6) Discussions:

- We revised the introductory paragraph, as suggested.

- To date, no data about SARS-COV2 and DOC, were reported.

- We explained implications of the study.

- We better explained limitation of the study design.

---

## [Decision Letter · Decision Letter 1]

30 Mar 2021

PONE-D-20-26757R1

The impact of the SARS-COV2 infection on the Disorder of Consciousness Rehabilitation Unit

PLOS ONE

Dear Dr. Ciurleo,

Thank you for submitting your manuscript to PLOS ONE. After careful consideration, we feel that it has merit but does not fully meet PLOS ONE’s publication criteria as it currently stands. Therefore, we invite you to submit a revised version of the manuscript that addresses the points raised during the review process.

ACADEMIC EDITOR: I have received the comments of the reviewers on your manuscript. The specific comments of the reviewers are included below. Please provide point by point response in your revised manuscript.

We look forward to receiving your revised manuscript.

Kind regards,

Muhammad Adrish, MD, MBA, FCCP, FCCM

Academic Editor

PLOS ONE

Journal Requirements:

Reviewers' comments:

Reviewer's Responses to Questions

**Comments to the Author**

1. If the authors have adequately addressed your comments raised in a previous round of review and you feel that this manuscript is now acceptable for publication, you may indicate that here to bypass the “Comments to the Author” section, enter your conflict of interest statement in the “Confidential to Editor” section, and submit your "Accept" recommendation.

Reviewer #1: All comments have been addressed

Reviewer #2: (No Response)

2. Is the manuscript technically sound, and do the data support the conclusions?

Reviewer #1: Yes

Reviewer #2: Partly

3. Has the statistical analysis been performed appropriately and rigorously? 

Reviewer #1: Yes

Reviewer #2: Yes

4. Have the authors made all data underlying the findings in their manuscript fully available?

Reviewer #1: Yes

Reviewer #2: Yes

5. Is the manuscript presented in an intelligible fashion and written in standard English?

Reviewer #1: Yes

Reviewer #2: Yes

6. Review Comments to the Author

Reviewer #1: Thank you for this longitudinal small-scale study. I would like the inclusion of the following in the retrospective paper: the steroid regimen use and dose regimen in the 18 patients, the body mass index for the 18 patient as it has been proven that an increased BMI contributes to increased mortality related to COVI-19 chest infection, and finally the inclusion of the metabolic profile including glycemic state control, electrolytes and BP control as all these could contribute to the encephalopathy cycle state of the patients.

Reviewer #2: Appreciating the authors for address the required corrections. However, some minor corrections should be addressed:

- I suggest authors add "pilot" to the design of the study in accordance with the small sample size.

- The methods section should be re-framed and re-arranged as the following:

a. Study design, setting

b. Participants

c. Intervention/issue of interest (exposure)

d. Comparison

e. Ethics and endpoint

f. Statistical analysis

- Clarify what the statistical method used to assess the data normality.

- Cite the Statistical Software used in data analysis.

7. PLOS authors have the option to publish the peer review history of their article (what does this mean?). If published, this will include your full peer review and any attached files.

Reviewer #1: **Yes: **MM

Reviewer #2: **Yes: **Walid Kamal Abdelbasset

---

## [Author Response · Author response to Decision Letter 1]

22 Apr 2021

Dear Editor, 

detailed responses to the reviewers’ questions appear below:

Reviewer #1: 

We thank the reviewer for the kind and appreciative words. We revised the paper, as suggested.

 - we included steroid regimen use;

 - we included BMI;

 - we included metabolic profile.

Reviewer #2: 

We thank the reviewer for the time spent to revise the manuscript. We revised the paper, as suggested.

Statistical section, already reports what was requested by the reviewer:

 - statistical method used: "Mean and standard deviations (SD) were used for normally distributed data and median and range for data that were not normally distributed. Continuous variables were compared by using Wilcoxon rank sum test. Proportions for categorical variables were compared using χ2 test. Because of the small sample size, ordinal, interval and ratio measures are presented as median and range";

 - statistical software used: "All statistical analyses were performed using R (version 3.3)".

We thank you again for your consideration of our work and look forward to hearing from you in due course.

Sincerely,

Rosella Ciurleo PharmD, PhD 

IRCCS Centro Neurolesi “Bonino-Pulejo”

S.S. 113, Via Palermo, C.da Casazza

98124 Messina, Italy

Phone: + 39-090-60128109

Fax: + 39-090-60128850

e-mail: rossella.ciurleo@irccsme.it

---

## [Decision Letter · Decision Letter 2]

11 May 2021

PONE-D-20-26757R2

The impact of the SARS-COV2 infection on the Disorder of Consciousness Rehabilitation Unit

PLOS ONE

Dear Dr. Ciuerlo,

Thank you for submitting your manuscript to PLOS ONE. After careful consideration, we feel that it has merit but does not fully meet PLOS ONE’s publication criteria as it currently stands. Therefore, we invite you to submit a revised version of the manuscript that addresses the points raised during the review process.

ACADEMIC EDITOR: Please review the edit suggested by the reviewer prior to acceptance. 

We look forward to receiving your revised manuscript.

Kind regards,

Muhammad Adrish, MD, MBA, FCCP, FCCM

Academic Editor

PLOS ONE

Journal Requirements:

Reviewers' comments:

Reviewer's Responses to Questions

**Comments to the Author**

1. If the authors have adequately addressed your comments raised in a previous round of review and you feel that this manuscript is now acceptable for publication, you may indicate that here to bypass the “Comments to the Author” section, enter your conflict of interest statement in the “Confidential to Editor” section, and submit your "Accept" recommendation.

Reviewer #1: All comments have been addressed

Reviewer #2: All comments have been addressed

2. Is the manuscript technically sound, and do the data support the conclusions?

Reviewer #1: Yes

Reviewer #2: Yes

3. Has the statistical analysis been performed appropriately and rigorously? 

Reviewer #1: Yes

Reviewer #2: Yes

4. Have the authors made all data underlying the findings in their manuscript fully available?

Reviewer #1: Yes

Reviewer #2: Yes

5. Is the manuscript presented in an intelligible fashion and written in standard English?

Reviewer #1: No

Reviewer #2: Yes

6. Review Comments to the Author

Reviewer #1: I thank the authors for considering the review feedback. Kindly revise the grammatical context for there are subtle subject-verb agreement errors such as “ The male patient did not presented fever, had normal white blood cell counts and he did not showed typical lung CT findings for COVID-19 infection “

Reviewer #2: (No Response)

7. PLOS authors have the option to publish the peer review history of their article (what does this mean?). If published, this will include your full peer review and any attached files.

Reviewer #1: **Yes: **MM

Reviewer #2: No

---

## [Author Response · Author response to Decision Letter 2]

23 May 2021

We regret there were problems with the English. The paper has been carefully revised to improve the grammar and readability.

We have highlighted the changes in yellow. The words or sentences deleted are in red.

---

## [Editor Report · Decision Letter 3]

1 Jun 2021

PONE-D-20-26757R3

The impact of the SARS-COV2 infection on the Disorder of Consciousness Rehabilitation Unit

PLOS ONE

Dear Dr. Ciurleo,

Thank you for submitting your manuscript to PLOS ONE. After careful consideration, we feel that it has merit but does not fully meet PLOS ONE’s publication criteria as it currently stands. Therefore, we invite you to submit a revised version of the manuscript that addresses the points raised during the review process.

ACADEMIC EDITOR: Please provide necessary changes as suggested below

We look forward to receiving your revised manuscript.

Kind regards,

Muhammad Adrish, MD, MBA, FCCP, FCCM

Academic Editor

PLOS ONE

Journal Requirements:

Additional Editor Comments (if provided):

I thank the authors for considering the review feedback. Kindly revise the grammatical context for there are subtle subject-verb agreement errors such as “ The male patient did not presented fever, had normal white blood cell counts and he did not showed typical lung CT findings for COVID-19 infection “

Reviewers' comments:

I thank the authors for considering the review feedback. Kindly revise the grammatical context for there are subtle subject-verb agreement errors such as “ The male patient did not presented fever, had normal white blood cell counts and he did not showed typical lung CT findings for COVID-19 infection “

---

## [Author Response · Author response to Decision Letter 3]

5 Jun 2021

Reviewer: 

We thank the reviewer for the kind and appreciative words. We revised the paper, as suggested.

 - we revised grammatical context as required.

---

## [Decision Letter · Decision Letter 4]

17 Jun 2021

The impact of the SARS-COV2 infection on the Disorder of Consciousness Rehabilitation Unit

PONE-D-20-26757R4

Dear Dr. Ciurleo,

We’re pleased to inform you that your manuscript has been judged scientifically suitable for publication and will be formally accepted for publication once it meets all outstanding technical requirements.

Kind regards,

Muhammad Adrish, MD, MBA, FCCP, FCCM

Academic Editor

PLOS ONE

Additional Editor Comments (optional):

All comments have been addressed.

Reviewers' comments:

Reviewer's Responses to Questions

**Comments to the Author**

1. If the authors have adequately addressed your comments raised in a previous round of review and you feel that this manuscript is now acceptable for publication, you may indicate that here to bypass the “Comments to the Author” section, enter your conflict of interest statement in the “Confidential to Editor” section, and submit your "Accept" recommendation.

Reviewer #1: All comments have been addressed

2. Is the manuscript technically sound, and do the data support the conclusions?

Reviewer #1: Yes

3. Has the statistical analysis been performed appropriately and rigorously? 

Reviewer #1: Yes

4. Have the authors made all data underlying the findings in their manuscript fully available?

Reviewer #1: Yes

5. Is the manuscript presented in an intelligible fashion and written in standard English?

Reviewer #1: Yes

6. Review Comments to the Author

Reviewer #1: I thank the authors for their vigorous efforts both in manuscript preparation and abiding by the reviewers’ recommendations

7. PLOS authors have the option to publish the peer review history of their article (what does this mean?). If published, this will include your full peer review and any attached files.

Reviewer #1: **Yes: **MM

---

## [Editor Report · Acceptance letter]

21 Jun 2021

PONE-D-20-26757R4 

The impact of the SARS-COV2 infection on the Disorder of Consciousness Rehabilitation Unit 

Dear Dr. Ciurleo:

I'm pleased to inform you that your manuscript has been deemed suitable for publication in PLOS ONE. Congratulations! Your manuscript is now with our production department. 

Kind regards, 

on behalf of

Dr. Muhammad Adrish 

Academic Editor

PLOS ONE